# Proposal of a guide for the interpretation, simplification of the regulatory process and good tax compliance, case of digital taxpayers, influencers

**Karen Yosio Mamani Monrroy**[1], **Nelly Rosario Moreno-Leyva**[1], **Kodi Santander**[1], **Shirley Eliza Salinas**[1], **Jorge Sánchez-Garcés**[1,2]*

**1** Escuela Profesional de Contabilidad, Facultad de Ciencias Empresariales, Universidad Peruana Unión, Puno, Perú, **2** Facultad de Ingeniería y Gestión, Universidad Nacional Tecnológica de Lima sur, Lima, Perú

* jsanchezg@untels.edu.pe

## Abstract

Influencers generate opinions in individuals through multiple virtual platforms, this phenomenon implies social influence that induces consumers to buy and direct these activities to the sponsorship of brands, which means monetary income for the influencer. Many of these incomes are not reported to the tax system, which causes evasion due to misinformation or lack of knowledge. Therefore, the need for a correct adaptation and interpretation of the Peruvian tax regulations for the payment of taxes on income received by this segment of taxpayers was observed. The purpose of this research was a guide that interprets, simplifies the processes and provides a regulatory framework for tax compliance for domiciled and non-domiciled influencers. The tax guide was designed thanks to the adaptation of the Scribber methodology and consisted of 4 steps: Familiarization, coding, theme generation, defining themes. The guide was organized in level 01, describing how to achieve the tax obligation in the sector of digital taxpayers influencers, level 02, where the activities described by the regulations are mentioned and level 3, tax procedures carried out by the tax administration to influencers. This guide is an aid to define the category that attributes the taxpayer's tax payment method. By identifying the tax categorization code according to the type of activity. It identifies the key factors to be able to interpret and adapt the law to the influencer's activities.

## Introduction

[1] describes that due to the decrease in the effectiveness of traditional advertising, brands have used influencers to spread their advertising, this activity is profitable due to the great diffusion that they generate with the target audience. [2] address the issue of influencer advertising, studying how brands influence their followers, the target audience for advertising. [3] indicate that persuasion is a phenomenon of social influence, being necessary to count on social networks to make it happen, [4] mention that social networks are communication

**Funding:** The authors received no specific funding for this work.

**Competing interests:** The authors have declared that no competing interests exist.

channels that allow free interaction, specialists reach consumers and interact with them. [5] allows the influencer to sponsor brands. [6] comment that influencers being drivers in sales induce the consumer to buy a product, making this sponsorship a monetary income to the influencer. [7] indicate about the Twitch platform, where influencers receive payments for live broadcasts; [8] adds about other payments for views on social networks when the video goes viral on the network. The income in this special segment of taxpayers (influencers) are mostly those that are not declared to the tax system and generate non-compliance with tax obligations. [9] mention that many of the influencers prioritize tax compliance and their tax obligations, making their followers follow the same behavior, because the social norm mentions that their individual tax behavior will depend on the expected net monetary benefit, tha's if tax evasion is going to harm their social reputation, then the influencer will take it very seriously to regularize their income to the tax system. On the other hand according to [10] the cause could be the lack of adequate instructive reports on tax issues towards the taxpayer. [11] corroborate that these reports are ineffective in making influencers aware of their tax obligations. According to [12]. this problem is compounded by the need for effective auditing and auditing in the absence of tax pressure, which leads to taxpayer non-compliance. [13] indicates that the state must carry out an update in its fiscal control processes, this allows the tax administration the support to refute statements where the content generators argue I don't understand, I didn't know.

Table 1 shows results of studies related to taxes for the provision of e-commerce/services. The antecedents refer tax avoidance of e-commerce companies according to [12], due to high tax rates for transaction costs, argue [14]. The urgent need to promote the tax ethics of e-commerce taxpayers, indicate [20] and the fiscal response to face the digital economy, tax treatment in relation to digital operations, effects of the rules proposed for e-commerce platforms to record the tax, as indicated by [18, 19].

The background described above studies the problem posed, however, they don't elaborate a guide to formalization and to comply with the payment of taxes on the income obtained. The need for a correct adaptation and interpretation of the Peruvian tax regulations for the payment of taxes on the income received by this segment of taxpayers was observed. In this sense, this research analyzed the regulatory framework and tax procedures in order to propose a guide that interprets and simplifies the formalization and tax compliance of domiciled and non-domiciled influencers. This guide was created thanks to the formulation of questions, which were answered in the results section and the discussion section. What would be the procedure for this sector of taxpayers to belong to the formal tax system? What are the taxes to which an influencer is obliged to pay and how to determine it? How does the tax administration control the compliance of this group of taxpayers with their obligations?

## Materials and methods

### Sample

This study was evaluated and approved by the Research Ethics Committee of the Universidad Peruana Union University, considering its scientific quality, the consideration of the welfare of its participants and compliance with the ethical norms established in the Code of Ethics for Research at Universidad Peruana Unión. the Universidad Peruana Unión. The sample consisted of 13 participants: 10 influencers and 3 tax experts.

- Inclusion:
  Influencers have a digital business, are public people on social networks, content creators and have an average number of 500,000 followers in social networks.

**Table 1. Representative publications that address the issue of egulatory process and good tax compliance, case of influencers from different solution approaches.**

| Proposal | Techniques | Results | Ref. |
|---|---|---|---|
| To highlight the unfavorable effects of labor social security on e-commerce caused by tax avoidance | Empirical research methodology, using comparison procedures and sensitivity analysis with samples of companies engaged in e-commerce. | Showed that there is greater labor tax avoidance in e-commerce companies compared to traditional ones. | [12] |
| Analyzes the effect of different tax jurisdictions on countries with high and low tax rates and how it affects their tax revenues. | Linear Spatial Hotelling Models | E-commerce and tax revenues in countries are affected by tax rates due to transaction costs | [14] |
| How influencers' income should be taxed | Structured interview. formal dogmatic method, rules of linguistic interpretation, comparative method. | No need to adopt new tax law rules specific measures to adequately tax the income of influencers, at least in developed countries | [15] |
| Addressing the issue of fair taxation of the digital economy | Comparative analysis | Allocate the tax equally to a traditional and a digital business, thus reducing tax inequality | [16] |
| Monetary structure with online payments and virtual goods | Algorithm test (Pow) | Increased growth of virtual currency compared to digital currency | [17] |
| It studies the global tax response to the digital economy in terms of taxation. | Compilation of evidence | The tax measures were acquired at the beginning of the proposed reform, however, they were not permanently adapted due to their inconsistency | [18] |
| Recognize the effects of the rules proposed for e-commerce platforms to record the tax | Comparative analysis | The measures employed increased the administrative burden as well as costs, noting that these rules are far from perfect | [19] |
| Demonstrate that e-commerce generates more tax avoidance | Empirical analysis | It is found that companies that do not engage in e-commerce are less evasive than those that do engage in this activity | [20] |
| provide a critical view of possible tax changes at the international level with respect to the challenges posed by the digitization of the economy | Bibliographic analysis | Introducing taxes to replace international tax treaties, so that each country can thus tax non-resident digital companies. | [21] |
| Offers an answer to the taxation of influencers' income if it were not properly regulated. | Bibliographic analysis | It would bring problems to identify their tax and tax payment, tax expenses to countries. | [11] |
| Reflects on new provisions to ensure oversight of the digital economy | Bibliographic analysis | The tax changes have created new complexities and challenges of tax interpretation for companies and for the customs authority | [22] |
| Consider how a country with an underdeveloped tax system can meet the challenges of digital commerce. | Doctrinal approach | There is a need for strategies that can be implemented in the institutions in order to create a tax regime suited to the digital era | [23] |

Experts have experience in tax areas, dedicated to legal and tax advisory services for companies, accountants, etc.

- Exclusion: Basically, it consists of discriminating professionals who are not in the Peruvian tax field; as for influencers, people who do not have a significant number of followers were not considered despite the fact that they performed work of publishing digital content on social networks.

## Scribber method

[24] describe the method that allows analyzing the information in the interviewees' responses, giving emphasis to the interviewees' narratives, This analysis is described in Fig 1 and consists of 5 steps.

- Step 01, Familiarization: the researcher transcribes the audio and video recordings of the interviews and then reads the texts of the responses in order to become familiar with the information recorded

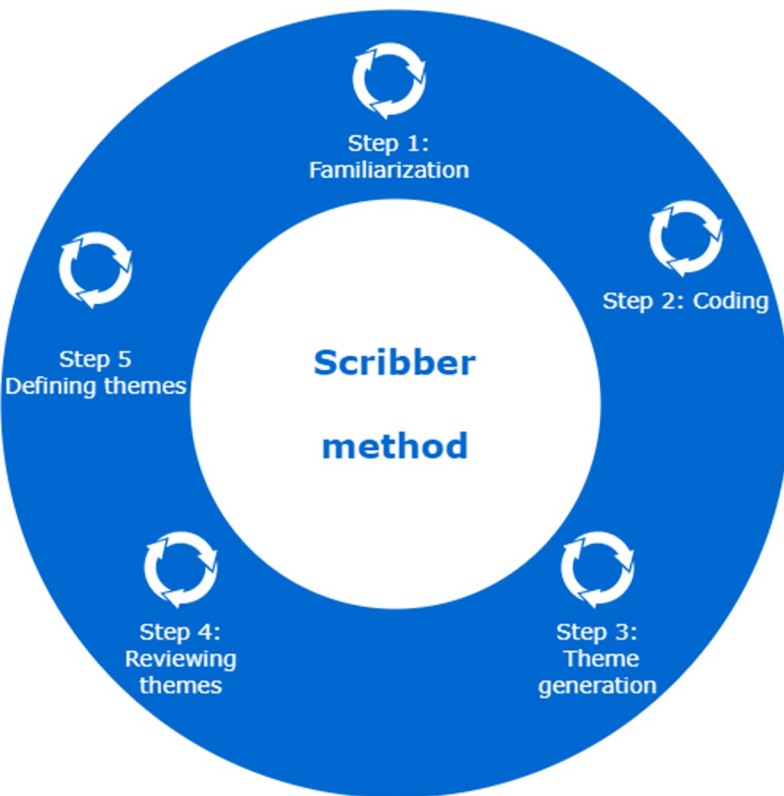

**Fig 1. Description of the proposed methodology.**

- Step 02, Coding: the coding of the information consists of highlighting sections of the text with key phrases, which are highlighted because they identify important aspects to be used in the resolution of the problem.

- Step 03, Theme generation: key phrases are grouped into categories called themes. These groupings define concepts, similar characteristics and identify patterns that explain possible solutions to problems.

- Step 04, Reviewing themes: a feedback of step 03 should be done, so that the identified categories are useful and accurate. For this we compare the themes with the data set; if we find any problems with the identified themes, they can be divided, combined, discarded or new ones created.

- Step 5: Defining themes: this step consists of conceptualizing, giving meaning to each group of codes (theme) involves formulating exactly what we mean by each theme and discovering how it helps us to understand the data and answer the research questions.

## Methodology to research

This section describes the adaptation of the steps of the Scribber methodology according to the needs of the research, as illustrated in Fig 2.

- Step 01, Familiarization: the interview guide was prepared according to Table 2, with the questions addressed to the experts and the influencers. The responses were transcribed and

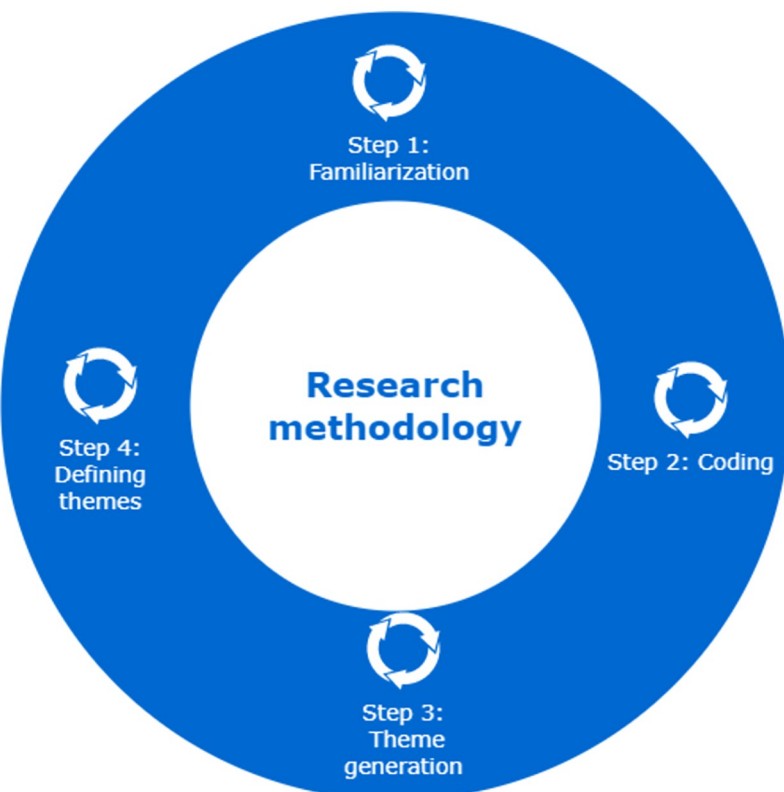

**Fig 2. Adaptation of the Scribber method to the research methodology.**

then analyzed for the purposes of the research.

This is the same step as Scribber

For the application of the interview guide, semi-structured questions were used, which made it possible to formulate new questions necessary for the interviewer to fully understand what the interviewee said.

- Step 02, Coding: For coding, step 2 of Scribber was used. Coding the information consists of highlighting sections of text from the responses of experts, influencers and regulations and recognizing them as key phrases to identify important aspects, and then relating the key phrases to meanings that will translate or interpret the key words into words known in the researcher's professional background. These meanings are called codes. Thanks to the coding, the procedure for registration in the tax register of the digital taxpayers sector

**Table 2. Interview guide, contextualization of the tax regulations.**

| Interviewee | Description of questions | Number of questions |
|---|---|---|
| Tax expert | *Tax collection through the application of regulations | 1 |
| | *Recommendations for the regulatory framework in the case of influencers | 1 |
| | *Recommendations to avoid evasion | 1 |
| Influencers | *The trajectory in social networks | 6 |
| | *Influencer activity, and the consequence it generates | 11 |
| | *Tax description of your activities | 9 |

(influencers) was obtained, then the activities that influencers perform to generate income were identified, and finally the tax procedures that they must comply with for the declaration and payment of their taxes. To perform this task, a table was used with columns such as topic, subtopic, key phrase, author of the phrase. Here were located the background elements that were the basis for the formulation of the tax guide. The details can be found at https://github.com/jasg1612/influencers, S1 File.

- Step 03, theme generation In this step, similar characteristics among the codes are analyzed in order to be grouped into themes and to describe important findings for the analysis. After grouping, characteristics could be identified and procedures could be defined. In the URL table https://github.com/jasg1612/influencers, S1 File the, themes and sub-themes are listed.

- Step 04: Defining themes:
  This step allows us to conceptualize, give meaning to the topics based on the meanings (codes) identified in the group of codes and formulate exactly what we understand in order to subsequently define the tax guide. According to the above, the topics were:
  **Formalization**, the formalization processes (choosing the type of RUC, registering in the RUC, choosing the income category and/or tax regime, obtaining the SOL code, issuing the payment voucher according to income category) were defined thanks to the interviews with experts and influencers and to the tax regulations that contain the formalization procedures according to the tax entity (SUNAT).
  **Tax determination**, the diversity of digital income identified in the interviews with influencers was contrasted with the tax regulations, thus limiting taxes and obligations for the activity of influencers, to then address the extensions with income tax and IGV for content creators.
  **Audit**, the tax violations found in the interviews with the experts were identified by means of the tax regulations with respect to auditing, and the new measures taken by the tax collection agency for this digital activity, thus touching on tax inducement, violations and penalties, tax crime.
  Then, the interpretation of the tax guide was performed. The table at https://github.com/jasg1612/influencers, S1 File; describing the code groups (topics) was used to elaborate the procedures for each topic identified in step 3. Each procedure was illustrated in Fig 3 of S1 Appendix. This simplifies the taxpayer registration procedure, the categorization of taxpayers based on their income and, finally, the tax formalities they must comply with to file and pay their taxes. Therefore, it is determined how the tax obligation should be reached. The analysis was accompanied with the AtlasTi software, to illustrate the occurrences between the codes of the key phrases and the answers of the interviewees, obtaining the Sankey figures, which illustrated the participation of the interviewees to obtain the tax guide.
  The research [25] is an example of how Sankey diagrams are used to explain relevant factors illustrated by colored arrows that are thicker than others, meaning that some code groups were more predominant and therefore were commonalities between code groups and respondents' opinions. Scribber step 04 and 05 were used to define this step.

## Results

The tax guide is detailed in S1 Appendix through the Fig 3;; which was developed in categories; the first category is "Formalization", it describes step 1 which corresponds to the choice of the type of taxpayer, where a process of individual incorporation or as a company is shown. Step 2 details the registration in the Single Taxpayers Registry (RUC), which is the registration in the

tax administration system. Step 3 specifies the choice of income category and/or tax regime to establish formal and substantial obligations, submitting information on expenses, income, payment of taxes within the established deadlines. According to tax expert 001: "according to the Influencer's activities, the Influencer must choose the type of tax regime and then issue the vouchers". This quote was used to obtain the codes tax obligations according to tax regime, issuance of vouchers according to tax regime; shown in Fig 4 in S2 Appendix; it describes the tax obligations of this sector of taxpayers; supported by the Payment Vouchers Regulation according to Superintendence Resolution Nº 007–99/SUNAT.

Step 4, indicates the access to the online tax system through the key of the Online Operations System (SOL), called SOL portal with the use of the RUC. This step is endorsed by Superintendence Resolution Nº 109–000/SUNAT and allows the issuance of payment vouchers. Step 5, explains how to issue the payment voucher according to income category, each taxpayer has the obligation to issue payment vouchers that accredit the income obtained. Tax expert 001 mentioned: "Every physical and/or virtual business activity or any option to session an asset has the obligation to be formalized" with this quote was obtained the code Every business is obliged to be formalized shown in Fig 4 of S2 Appendix, highlighting the importance of the activity of the formal taxpayer; the five steps were illustrated in S1 Appendix.

The next category was the "Determination of the tax" which consists of the calculation of the tax based on the income obtained, which may be of domestic and/or foreign origin. Now, to the extent that the origin is determined to be of national source, the Influencers are obliged to calculate two taxes: General Sales Tax (IGV) and Income Tax (IR) and in case it is considered that the income is of foreign source, the Influencers would be in the obligation to calculate only the Income Tax, established in S1 Appendix attached.

Income Tax in Peru is the tax that levies a certain percentage of payment on the income or profits obtained, which can be defined in the following categories: capital income, labor income or business income. Tax expert 002 expresses: "it would be necessary to make an analysis on the type of taxes, type of regimes, types of formal and substantial obligations to which it is obliged, definitely here there is no issue of incorporation to the norm, the Influencers are already included in the tax regimes", from this quote was obtained the code need of tax analysis, shown in the Fig 5 of S2 Appendix.

The statement highlights the importance of analyzing the activity of each Influencer and knowing the context of how the income is generated in order to establish the forms of payment of their taxes on a Peruvian standard already determined; that is, the activity of the Influencer does not need to be incorporated but identified in the legal basis in terms of income already determined.

The categories are defined by previously identifying whether the income comes from an asset, which is transferred and/or exploited for the generation of income; this income is called capital income. On the other hand, there is labor income, in this case, the income is generated by the rendering of personal services, which are reflected in salaries, fees and/or commissions. However, it should be noted that Peruvian law recognizes a third type of income called business income, which is defined as income generated by the combination of capital plus labor. This analysis is important because it helps to define the category that attributes the form of payment of taxes. Identifying whether the income comes from the exploitation of a capital, from the rendering of its services or from the combination of both; depending on the type of activity. As for the Influencers and according to the analysis made in the diagram, they would only be involved in business income or labor income but not in capital income. In the attached diagram 01, business income is presented under the denomination of third category income, which contemplates several regimes, of which an Influencer, due to the diversity of activities he/she performs, could only consider the Mype regime or the general regime, in both cases he/

she would pay two taxes: IGV and IR. If the annual income does not exceed 1700 tax units (UIT), the Influencer could join the MYPE regime, in which he/she would have to pay 18% for IGV and a progressive rate of 10 to 29.5% for IR. On the other hand, if the annual income exceeds 1700 UIT, the Influencers' option is the general regime in which they pay 18% for IGV and 29.5% for IR. This percentage is applied on the economic profit obtained by the difference of their income minus their expenses within the period of one year. As for the UIT, it is a reference tax value in Peru that varies each year according to macroeconomic conditions; for example, in 2021 it was 4,400 soles and in 2022 it was 4,600 soles.

The labor income in the diagram, S1 Appendix is shown in two forms: fourth category income that represents the independent work of the Influencer and fifth category income that represents the work of the Influencer under a dependency relationship with an employer.

The analysis according to the activity of the Influencers is exercised by the provision of their independent services, such as those detailed in Influencers 008; that is, own activities without the subordination of a third party, without establishing schedules and without taking into account the capital. This comment arose when indicating the types of income received as an influencer due to the monetization in the web platform where he performs his activities, the key phrases were monetization by activity, monetization by advertising, monetization in social networks, monetization of content shown in Fig 6 of S2 Appendix. This type of income in Peru presents a progressive tax burden under a scale of: 8%, 14%, 17%, 17%, 20% and 30% on net income. The net income is calculated by the difference of the total income received minus two deductions; one of 20% and the other of 7 UIT's.

On the other hand, the legal framework contemplates the recognition of fifth category income within labor income, but under the condition of dependency, that is, that the Influencer's work is subordinated by a third party, which in most cases can be a marketing agency or a third party that hires their services. As an example is what was explained by Influencers 010 who expressed: "I have an agreement, a contract with a schedule, which asks me certain requirements and when I meet them I have a paid job", in this quote the influencer points out the characteristics of a dependent job so the phrase working conditions shown in Fig 7 of S2 Appendix, it is pointed out that there is a dependent contract where the schedules and standards that must be met for their remuneration based on Article 34 of the income tax law are established. This group of Influencers maintain a subordinate contract, the tax rate is the same as independent services; 8%, 14%, 17%, 20% and 30% on net income. However, the difference is that the net income is calculated by the difference of the total income received minus only a deduction of 7 UIT.

The third part of the guide is comprised by the category of "Auditing", which is developed considering the following aspects: inducement, infractions and penalties; also incorporating criteria related to tax crimes and/or tax fraud.

The audit process is the stage supported by article 62 of the TUO of the Tax Code that sustains the power to audit taxpayers, it seeks to regularize their stage of formalization, regularization and/or determination of tax payment. It identifies the presumed income of the activities of the Influencers, In turn, tax expert 003 points out that the identification of the income in most cases is obtained based on the ITF (Tax on Financial Transactions) for which the key phrase taxation was produced through the ITF shown in Fig 8 of S2 Appendix which is an informative tax on all the financial operations executed by a taxpayer, giving the Treasury a reference of the amounts generated by his work. If the Tax Administration corroborates that such incomes were not communicated or were partially communicated to the Treasury, it proceeds to carry out an inductive work in which it induces the Influencers to regularize the payment of the tax for the detected incomes under the presumption of "Unjustified Patrimony".

Finally, the aspect of "Penalties and fines" is presented once the taxpayer, even with all the induction stages, does not regularize the payment of his taxes or does it erroneously, out of ignorance, i.e., without fraud. This forces the tax administration to establish penalties and fines. These are represented by pecuniary payments, such penalty scale is based on the level of non-compliance. In case the non-compliance is made with "malice", that is, with intent to violate the law, the act is considered a "tax crime", which is punishable by imprisonment.

Therefore, the tax guide has been formulated based on the opinions of experts and Influencers, the laws and regulations of the Peruvian tax system; obtaining a series of codes (key phrases) that allowed to illustrate all the components of the tax guide.

## Discussions

Discussions have been conducted on the basis of the questions posed in the introduction, discussing each finding found in the design of the fiscal guide through the relevant answers from tax experts and influencers, highlighting these findings relevant to the process of formalization of influencers (digital taxpayers). 1. What would be the procedure to incorporate this sector of taxpayers into the formal tax system? Part 1 of the diagram shows a category called "Formalization", which shows all the steps to follow according to the tax regulations to formalize and thus obtain an identified tax that allows the registration in the single taxpayer registry (RUC), then choose a category depending on the activities performed and thus comply with their obligations to the Treasury, although the Peruvian regulations do not refer in explicit terms to "Influencers", but request a process of interpretation that in most cases identifies it as a business activity. [15] agrees that it is not necessary to change the regulations to tax influencers, but it is necessary for the tax administration to identify them, which allows finding a solution by informing this economic sector with a guide that allows them to be integrated into the tax system.

[23] they state that digital activities, advertising on networks or e-commerce interaction, is subject to taxation according to tax regulations, so their formalization process is done only through a corporate figure; which is partially related to the Peruvian legislation. On the other hand, [18] point to countries whose tax legislation still poses problems when identifying the digitalized economy, a fact that causes difficulties in its formalization and the tax collection process. At the international level [11] indicates that the digital creation of a person acting as an influencer is thanks to an artificial intelligence that makes profits from the influencer's activity for the agency that created it. According to art. 7 of the OECD model and the UN model these activities are considered as business income.

2. What taxes do influencers have to pay and how to determine them?

The second category of the tax guide shows the process of "Tax Determination", which indicates as a primary step the identification of the domiciled or non-domiciled (foreign) source of income, in order to reference the necessary tax payments. It also explains how to choose your income category, mentions the admissible tax regimes depending on the activity you perform. A requirement at the moment of choosing the regime, as indicated by [26], is to first identify the tax regulations for the application of the tax in the corresponding tax category.

[15] agrees with the research that the analysis of the activity of the influencers should be taken into account in relation to their income, in order to incorporate their tax obligations. [27] developed in China a digital tool that analyzes the content and information of a publication made by the influencer, if it 287 performs marketing, the income is imposed on the necessary obligations for the taxpayer.

However, [28] considers that the taxation of the digital economy should not be considered by its activity, he also indicates that the tax process of this type of taxpayers should not compromise their profitability, but on the contrary their contribution should be made according to their ability to pay; agreeing with [21] in an international context, the creation of value is taken into consideration for the allocation of taxes and not the type of source.

3. How does the tax administration monitor the compliance of this group of taxpayers with their obligations?

The audit process points out the inducements carried out by the tax administration to capture the digital economic sector (influencer), and the economic sanctions or seizures of goods for non-compliance with tax regulations, in the event of non-compliance due to ignorance of the tax regulations without the intention of a fraud process; otherwise, if fraudulent intent to defraud is demonstrated, is considered a criminally punishable offense. [29] addresses the issue of tax fraud under a perspective that tax morale would be composed by the trust of the tax system towards taxpayers [18] studied the global tax response to the digital economy through evidence gathering, with the result that tax measures were acquired at the beginning of the proposed reform, however, the digital economy failed to adapt due to the ongoing inconsistency of these measures. The third category of the study is taxation. To date, the Peruvian tax administration has been developing actions to better control tax evasion by digital content generators in social networks. The results reflect that influencers need guidance to avoid future penalties or fines for tax evasion. [30] identifies tax consequences, such as fines or penalties, which it indicates could be combated by streamlining interaction with taxpayers through incentives.

[31] mentions that, in the future, taxation in the area of e-commerce will need to be more intense and therefore tax consequences will need to be more intense.

## Conclusions

For influencers, the tax obligations imposed by the tax administration by considering them all as entrepreneurs, may cause this group of taxpayers to stop paying taxes or evade them due to the tax burden and the obligations they must comply with as formal taxpayers. To this effect, induction and/or orientation campaigns for this group of income generating taxpayers should be taken into account. It is important to analyze the activities developed by Influencers to categorize them in the tax system, in order to avoid omissions and/or penalties. The tax administration is obliged to generate tax risk in social networks to address the fight against evasion in this economic group, and thus increase the collection in digital media.

Recently, the tax authority is taking control measures in social networks, thus evidencing its conditioning to globalization, and new changes are expected in the way of inducing Influencers to formalization. The legal framework must literally identify the Influencers; who under the current regulation would fit in third category income under a process of interpretation thus relating the following codes: analysis of activity performed, place of work, analysis of tax category and use of capital.

It is important to analyze each activity performed by Influencers; for example, if they record a video promoting a brand and publish it on their social networks, if a brand requests them to publish its banner or if they record a video for a brand by order of a marketing agency. This analysis helps to define the category of income attributed by the form of payment of the tax. Identifying whether the income comes from the exploitation of capital, from the provision of services or from the combination of both; according to the tax categorization code and according to the type of activity. To this work is added the proposal of the legal guide that identifies the key factors to be able to interpret and adapt the law to the activities of Influencers.

Therefore, the diversity of income and expenses generated by their various activities as content creators can be used to deduct their tax impositions, which must be controlled.

## Supporting information

**S1 File. English-Tax guide.**
(XLSX)

**S2 File. Interview for the influencers.**
(DOCX)

**S3 File. Interview for the tax expert.**
(DOCX)

**S1 Appendix.**
(PDF)

**S2 Appendix.**
(PDF)

## Author Contributions

**Conceptualization:** Karen Yosio Mamani Monrroy, Nelly Rosario Moreno-Leyva, Shirley Eliza Salinas.

**Data curation:** Jorge Sánchez-Garcés.

**Formal analysis:** Karen Yosio Mamani Monrroy, Nelly Rosario Moreno-Leyva, Kodi Santander, Shirley Eliza Salinas, Jorge Sánchez-Garcés.

**Investigation:** Kodi Santander.

**Methodology:** Jorge Sánchez-Garcés.

**Project administration:** Jorge Sánchez-Garcés.

**Supervision:** Nelly Rosario Moreno-Leyva, Jorge Sánchez-Garcés.

**Validation:** Karen Yosio Mamani Monrroy, Nelly Rosario Moreno-Leyva, Kodi Santander, Jorge Sánchez-Garcés.

**Writing – original draft:** Karen Yosio Mamani Monrroy, Kodi Santander, Shirley Eliza Salinas, Jorge Sánchez-Garcés.

**Writing – review & editing:** Karen Yosio Mamani Monrroy, Nelly Rosario Moreno-Leyva, Kodi Santander, Jorge Sánchez-Garcés.

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
