## [Decision Letter · Decision Letter 0]

13 Mar 2023

PONE-D-22-35364Proposal of a guide for the interpretation, simplification of the regulatory process and good tax compliance, case of digital taxpayers, influencersPLOS ONE

Dear Dr. Sánchez-Garces,

Thank you for submitting your manuscript to PLOS ONE. After careful consideration, we feel that it has merit but does not fully meet PLOS ONE’s publication criteria as it currently stands. Therefore, we invite you to submit a revised version of the manuscript that addresses the points raised during the review process.

We look forward to receiving your revised manuscript.

Kind regards,

Jaime A. Yáñez, Ph.D.

Academic Editor

PLOS ONE

Journal Requirements:

Additional Editor Comments:

Dear Author,

Based on the peer reviews the recommendation is to perform major revisions before considering for acceptance.

Thanks, best regards

Academic Editor

Reviewers' comments:

Reviewer's Responses to Questions

**Comments to the Author**

1. Is the manuscript technically sound, and do the data support the conclusions?

Reviewer #1: Yes

Reviewer #2: Yes

2. Has the statistical analysis been performed appropriately and rigorously? 

Reviewer #1: Yes

Reviewer #2: Yes

3. Have the authors made all data underlying the findings in their manuscript fully available?

Reviewer #1: Yes

Reviewer #2: Yes

4. Is the manuscript presented in an intelligible fashion and written in standard English?

Reviewer #1: Yes

Reviewer #2: Yes

5. Review Comments to the Author

Reviewer #1: Thank you for the opportunity to make sugggestions to improve your manuscript.

1. Influencers are not opinion leaders. Please, review in deep the details to explain better the concept.

2. I think ithe aim must be an analysis. Make a guide NOT is a research outcome.

3. In methdology is mentioned "have a significant number of followers in social 46 networks". What means "significant number"? 1000 followers? 10 000? 100 000? one million?

3. I suggest process the data by ATLAS.TI or other qualitative software.

4. The topic is interesintng but need to improve the method of the study.

Reviewer #2: The title ("Proposal of a guide for the interpretation, simplification of the regulatory process and good tax compliance, case of digital taxpayers, influencers") and the summary present a correct structure, with an adequate identification of what is subsequently addressed in the text.

It is an article that contributes to society and helps to promote knowledge about the importance of digital taxpayers and their taxation, from the point of view of influencers.

The objectives and hypotheses are well stated and appropriate, with methodological rigour and well-designed and thought-out research instruments. Therefore, the results of the review are evident and relevant. The results are clearly presented, leading to a discussion section and conclusions that correlate with the objectives and methodology employed, as well as with the results obtained, all in a transparent and convincing manner.

It is a good article to which I would perhaps add some more global perspective and updated references. In this regard, I recommend citing the following references:

- DeMarzo, P., Vayanos, D., & Zwiebel, J. (2003). Persuasion bias, social influence, and unidimensional opinions. The Quarterly Journal of Economics, 118(3), 909– 968.

- Di Gioacchino, D., & Fichera, D. (2022). Tax evasion and social reputation: The role of influencers in a social network. Metroeconomica, 73(4), 1048-1069.

- Sanz-Marcos, P.; Jiménez-Marín, G.; Elías Zambrano, R. (2021). Aplicación y uso del Modelo de Resonancia o Customer-Based Brand Equity (CBBE). Estudio de la lealtad de marca a través de la figura del influencer. methaodos.revista de ciencias sociales 9 (2), 200-218

6. PLOS authors have the option to publish the peer review history of their article (what does this mean?). If published, this will include your full peer review and any attached files.

Reviewer #1: No

Reviewer #2: No

---

## [Author Response · Author response to Decision Letter 0]

12 Apr 2023

%%%%%%%%%%%%%%%%

% Reviewer 1 %

%%%%%%%%%%%%%%%%

1.Influencers are not opinion leaders. Please, review in deep the details to explain better the concept.

the change was made and the following was written in the first paragraph of the abstract and introduction: "Influencers generate opinions in individuals through multiple virtual platforms", using the author DeMarzo PM, Vayanos D, Zwiebel J. Persuasion Bias, Social Influence, and Unidimensional Opinions*. The Quarterly Journal of Economics. 2003;118(3):909–968. doi:10.1162/00335530360698469.

2. I think ithe aim must be an analysis. Make a guide NOT is a research outcome.

the change was made from line 33 to line 40 of the introduction, considering pertinent the reviewer's observation, since first an analysis of the tax regulatory framework was made and then the analysis of the experts' and influencers' answers.

3.In methdology is mentioned "have a significant number of followers in social 46 networks". What means "significant number"? 1000 followers? 10 000? 100 000? one million?

The phrase "have a significant number of followers" was specified in the sample inclusion section in materials and methods; in line 51 where an average number of followers among the 10 influencers of 500,000 is specified.

4.I suggest process the data by ATLAS.TI or other qualitative software.

The data analysis was performed by entering the documents of the answers for each respondent and their related codes using the AtlasTi software, obtaining in the analysis section of the software the occurrences of document, code and illustrating them in sankey diagrams of the software. These results were added in the results report, in annex 02 and their explanation in lines 95 to 104.

5.

he topic is interesintng but need to improve the method of the study

Once again, thank you very much for your pertinent comments in order to obtain a quality paper. 

The wording of the Scribber steps was improved, explaining in greater detail steps 01 (transcription of the videos), 02 (process of formulating meanings of the key phrases through qualitative coding), 03 (the themes defined through the group of meanings of the key phrases and that were identified as findings), 04 (explanation of the analysis carried out to obtain the tax guide as a product, accompanied with the data analysis of Atlas ti).

%%%%%%%%%%%%%%%%

% Reviewer 2 %

%%%%%%%%%%%%%%%%

1. The title ("Proposal of a guide for the interpretation, simplification of the regulatory process and good tax compliance, case of digital taxpayers, influencers") and the summary present a correct structure, with an adequate identification of what is subsequently addressed in the text.

It is an article that contributes to society and helps to promote knowledge about the importance of digital taxpayers and their taxation, from the point of view of influencers.

The objectives and hypotheses are well stated and appropriate, with methodological rigour and well-designed and thought-out research instruments. Therefore, the results of the review are evident and relevant. The results are clearly presented, leading to a discussion section and conclusions that correlate with the objectives and methodology employed, as well as with the results obtained, all in a transparent and convincing manner.

It is a good article to which I would perhaps add some more global perspective and updated references. In this regard, I recommend citing the following references

Authors were added to the introduction section, very important to improve the context of the research on lines 2, 9 and 10\\\\

- DeMarzo, P., Vayanos, D., & Zwiebel, J. (2003). Persuasion bias, social influence, and unidimensional opinions. The Quarterly Journal of Economics, 118(3), 909- 968.

\\\\

- Di Gioacchino, D., & Fichera, D. (2022). Tax evasion and social reputation: The role of influencers in a social network. Metroeconomica, 73(4), 1048-1069.

---

## [Decision Letter · Decision Letter 1]

22 May 2023

Proposal of a guide for the interpretation, simplification of the regulatory process and good tax compliance, case of digital taxpayers, influencers

PONE-D-22-35364R1

Dear Dr. Sánchez-Garces,

We’re pleased to inform you that your manuscript has been judged scientifically suitable for publication and will be formally accepted for publication once it meets all outstanding technical requirements.

Kind regards,

Godfred Matthew Yaw Owusu

Academic Editor

PLOS ONE

Additional Editor Comments (optional):

Authors should engage the services of professional proofreader to thoroughly read through the entire manuscript to correct all minor typos.

Reviewers' comments:

Reviewer's Responses to Questions

**Comments to the Author**

1. If the authors have adequately addressed your comments raised in a previous round of review and you feel that this manuscript is now acceptable for publication, you may indicate that here to bypass the “Comments to the Author” section, enter your conflict of interest statement in the “Confidential to Editor” section, and submit your "Accept" recommendation.

Reviewer #2: All comments have been addressed

2. Is the manuscript technically sound, and do the data support the conclusions?

Reviewer #2: Partly

3. Has the statistical analysis been performed appropriately and rigorously? 

Reviewer #2: Yes

4. Have the authors made all data underlying the findings in their manuscript fully available?

Reviewer #2: Yes

5. Is the manuscript presented in an intelligible fashion and written in standard English?

Reviewer #2: Yes

6. Review Comments to the Author

Reviewer #2: - I consider that influencers figures are not opinion leaders, although in many cases can be it. I suggest this term must be reviewed.

- The topic is so interesintng but need to improve the method of the study, maybe with a third method, impling a methodological triangulation.

-I would perhaps add some more and updated references, such as:

> Arnesson, J. (2022). Influencers as ideological intermediaries: Promotional politics and authenticity labour in influencer collaborations. Media, Culture & Society, 1–17. Advance online publication. https://doi.org/10.1177/01634437221117505

> Cross, S., & Littler, J. (2010). Celebrity and Schadenfreude. Cultural Studies, 24(3), 395–417. https://doi.org/10.1080/09502381003750344

> Giachi, S. (2014). Social dimensions of tax evasion: Trust and tax morale in contemporary Spain. Revista Española de Investigaciones Sociológicas, (145), 73–98. https://doi.org/10.5477/cis/reis.145.73

> Jiménez-Marín, G., Sanz-Marcos, P., & Tobar-Pesantez, L. B. (2021). Keller's resonance model in the context of fashion branding: persuasive impact through the figure of the influencer. Academy of Strategic Management Journal, 20 (6).

> Oliva, M., Tomasena, J. M., & Anglada-Pujol, O. (2023). ‘Kids, these YouTubers are stealing from you’: influencers and online discussions about taxes. Information, Communication & Society, 1-18.

> Sanz-Marcos, P., Jiménez-Marín, G., & Elías Zambrano, R. (2021). Aplicación y uso del Modelo de Resonancia o Customer-Based Brand Equity (CBBE). Estudio de la lealtad de marca a través de la figura del influencer. Methaodos. revista de ciencias sociales, 9 (2), 200-218.

7. PLOS authors have the option to publish the peer review history of their article (what does this mean?). If published, this will include your full peer review and any attached files.

Reviewer #2: No

---

## [Editor Report · Acceptance letter]

7 Jun 2023

PONE-D-22-35364R1 

Proposal of a guide for the interpretation, simplification of the regulatory process and good tax compliance, case of digital taxpayers, influencers 

Dear Dr. Sánchez-Garces:

I'm pleased to inform you that your manuscript has been deemed suitable for publication in PLOS ONE. Congratulations! Your manuscript is now with our production department. 

Kind regards, 

on behalf of

Dr. Godfred Matthew Yaw Owusu 

Academic Editor

PLOS ONE